# Neurosensory Rehabilitation and Olfactory Network Recovery in Covid-19-related Olfactory Dysfunction

**DOI:** 10.3390/brainsci11060686

**Published:** 2021-05-23

**Authors:** Tom Wai-Hin Chung, Hui Zhang, Fergus Kai-Chuen Wong, Siddharth Sridhar, Kwok-Hung Chan, Vincent Chi-Chung Cheng, Kwok-Yung Yuen, Ivan Fan-Ngai Hung, Henry Ka-Fung Mak

**Affiliations:** 1Department of Microbiology, Li Ka Shing Faculty of Medicine, The University of Hong Kong, Hong Kong, China; sid8998@hku.hk (S.S.); chankh2@hku.hk (K.-H.C.); chengccv@ha.org.hk (V.C.-C.C.); kyyuen@hku.hk (K.-Y.Y.); 2Department of Diagnostic Radiology, Li Ka Shing Faculty of Medicine, The University of Hong Kong, Hong Kong, China; zhanghuiok9108@163.com; 3Department of Ear, Nose and Throat Surgery, Pamela Youde Nethersole Eastern Hospital, Hong Kong, China; wongkaichuen@gmail.com; 4State Key Laboratory of Emerging Infectious Diseases, The University of Hong Kong, Hong Kong, China; 5Carol Yu Centre for Infection, The University of Hong Kong, Hong Kong, China; 6The Collaborative Innovation Center for Diagnosis and Treatment of Infectious Diseases, The University of Hong Kong, Hong Kong, China; 7Department of Medicine, Li Ka Shing Faculty of Medicine, The University of Hong Kong, Hong Kong, China; 8State Key Laboratory of Brain and Cognitive Sciences, The University of Hong Kong, Hong Kong, China; 9Alzheimer’s Disease Research Network, The University of Hong Kong, Hong Kong, China

**Keywords:** COVID-19, olfactory dysfunction, vitamin A, smell training, resting-state fMRI

## Abstract

Non-conductive olfactory dysfunction (OD) is an important extra-pulmonary manifestation of coronavirus disease 2019 (COVID-19). Olfactory bulb (OB) volume loss and olfactory network functional connectivity (FC) defects were identified in two patients suffering from prolonged COVID-19-related OD. One patient received olfactory treatment (OT) by the combination of oral vitamin A and smell training via the novel electronic portable aromatic rehabilitation (EPAR) diffusers. After four-weeks of OT, clinical recuperation of smell was correlated with interval increase of bilateral OB volumes [right: 22.5 mm^3^ to 49.5 mm^3^ (120%), left: 37.5 mm^3^ to 42 mm^3^ (12%)] and improvement of mean olfactory FC [0.09 to 0.15 (66.6%)].

## 1. Introduction

Non-conductive olfactory dysfunction (OD) is an important extra-pulmonary manifestation of coronavirus disease 2019 (COVID-19), which is caused by severe acute respiratory syndrome coronavirus 2 (SARS-CoV-2). Local experience and multiple cross-sectional studies revealed that up to 60% of COVID-19 patients suffer from OD, with a female predominance [1,2]. The severity and duration of COVID-19-related OD varies but olfactory chemosensory disturbances may persist beyond four weeks [3].

The neuroinvasive properties of SARS-CoV-2 at the olfactory epithelium (OE) has been confirmed in human autopsy specimens [4,5,6]. Furthermore, the neurotropic properties of SARS-CoV-2 have been demonstrated in multiple biological models, such as U251 human glioblastoma cell line, Tuj1^+^ Pax6^+^Nestin^+^ human brain organoids, and golden Syrian hamster models [7,8,9]. Direct invasion of SARS-CoV-2 at the delicate neural-mucosal interface between the olfactory and central nervous systems (CNS) may be the underlying pathophysiological cause of COVID-19-related OD.

OE is a pseudostratified columnar epithelium which consists of olfactory sensory neurons, non-neuronal supporting cells (for example: sustentacular cells, Bowman’s glands and ducts), as well as progenitor and stem cells. The OE is maintained continuously by globose basal cells (GBCs) throughout life. After substantial tissue injury, OE regeneration is further supported by the differentiation of multipotent horizontal basal cells (HBCs) [10,11]. Therefore, the regenerative potentials and neuroplasticity of OE may be harnessed in the treatment against COVID-19-related OD.

In this case report, one patient received oral vitamin A (VitA) in combination with smell training (ST) as olfactory treatment (OT) against prolonged COVID-19-related OD. VitA supplement was selected as a metabolic enhancer for OE regeneration based on previous studies [12,13]. In addition, the novel electronic portable aromatic rehabilitation (EPAR) diffuser (SOVOS, Hong Kong, China) technology was adopted for superior olfactory stimulation as an enhancement to conventional ST [14,15]. 

## 2. Case Presentation

In this case study, two patients diagnosed with prolonged COVID-19-related OD were recruited to undergo structural and resting-state functional MRI (rs-fMRI) brain scans. Two non-clinical staff were invited to participate as controls (Appendix A). COVID-19 patients received nasoendoscopic assessments. Detailed functional olfactory evaluations are outlined in Appendix A. SARS-CoV-2 virologic assessments can be found in Supplementary Note S1. 

All participants underwent MRI brain scans using a 1.5 T MR scanner (SIGNA; GE Healthcare, Chicago, IL, USA). Structural and three-dimensional (3D) arterial spin labeling images were acquired. Volumetric analyses of the olfactory bulbs (OB) were performed. MR spectroscopy was performed using the single voxel point resolved spectroscopy (PRESS) at the gyrus rectus (GR) and superior frontal cortex (SFC). 

rs-fMRI brain scans were collected using a gradient-echo echo-planar sequence sensitive to blood-oxygen-level-dependent (BOLD) contrast. Hypothesis-driven region of interest (ROI) approach was applied [16,17,18]. The olfactory network seed regions were defined at the caudate nuclei [19,20]. With reference to the automated anatomical labeling template, rs-fMRI data were segmented into 90 regions, of which 28 out of 90 regions were associated with the functional olfactory cortical networks (OCN; Figure 1). The functional connectivity (FC) between seed regions and OCN ROIs were obtained by extracting the average time series from individual ROIs and calculating the correlation. Detailed methodology of the MRI data acquisition can be found in Appendix A.

Serial rs-fMRI FC images of the olfactory network are shown (Figure 2A–C). Olfactory network FC were reduced in COVID-19 patients when compared to controls. FC were homogenously decreased in the right and left caudate seed regions for patient 1 (P1) and patient 2 (P2), respectively (Appendix A).

Structural MRI brain scans showed OB volume defects in COVID-19 patients (Appendix A) when compared to healthy controls. MR spectroscopy confirmed neuronal loss (Appendix A), as evident by reduction of *N*-acetylaspartate levels at the GR and SFC [21]. 

After baseline rs-fMRI assessments, OT was initiated for patient 1 (P1). Oral VitA 25,000 IU [7500 µg retinol activity equivalents (RAE); Carlson Laboratories, Arlington Heights, IL, USA] soft gels were prescribed daily for two weeks in combination with smell training (ST) thrice daily for four weeks via the novel EPAR diffusers. Methodological details of ST can be found in Appendix A.

Clinical improvements in olfaction were documented serially by subjective questionnaires (Appendix A) and objective olfactory quantitation (Appendix A) for P1 after OT. Butanol threshold test (BTT) revealed improvement in olfactory sensitivity from more than 4% to 1%. Smell identification test (SIT) confirmed categorical improvement in olfaction from anosmia to severe microsmia. Notably, there were measurable increase in the bilateral OB volumes [Appendix A; right OB: 22.5 mm^3^ to 49.5 mm^3^ (120%), left OB: 37.5 mm^3^ to 42 mm^3^ (12%)] in P1 after OT.

Olfactory recovery in P1 was correlated with significant improvements in the mean olfactory FC [0.09 to 0.15 (66.6%); Appendix A], when compared with pre-OT baseline scans. FC improvements were documented in both primary [left piriform cortex (PC), and right amygdala] and secondary (bilateral GR, and medial orbitofrontal cortices) OCN areas (Figure 2A–C). Importantly, there were corresponding increase in regional cerebral blood flow at the bilateral PC (left: 17.43%, right: 10.99%) and multiple secondary OCN regions (Appendix A), demonstrating robust neurovascular coupling.

## 3. Discussion

In this case report, COVID-19-related OD was correlated with OB volume loss and abnormal MR spectroscopy findings, indicating neuronal destruction in the CNS secondary to COVID-19 infection. Furthermore, rs-fMRI demonstrated olfactory FC impairments, thereby providing further insights into the underlying neuropathological process of COVID-19-related OD. 

In the treatment for prolonged COVID-19-related OD, P1 was successfully challenged with oral VitA and ST, as demonstrated by (1) interval improvements in olfactory function; (2) structural restoration in OB volumes; (3) olfactory network recovery; and (4) corresponding regional increase in cerebral perfusion. 

The long-term outcomes of patients suffering from prolonged COVID-19-related OD remain unknown [22]. Extended longitudinal follow-up studies will be needed to determine the prognosis of olfactory chemosensory deficits for these patients. The therapeutic efficacy of oral VitA and ST in the treatment against prolonged COVID-19-related OD should also be validated in large scale randomized–controlled trials, which would differentiate between spontaneous natural recovery and treatment cure.

The integrity of the mammalian OE is preserved by the mitotically active GBCs and dormant HBCs [10,11]. The dormancy of HBCs is maintained by Notch1 signaling which is correlated with the expression of transcription factor protein 63 (ΔNp63α) [23,24]. During extensive OE injury, Notch1 signals and ΔNp63α expressions are downregulated, leading to HBCs activation and differentiation. 

We hypothesize that VitA is an important metabolic substrate for robust neurogenesis at the olfactory apparatus, as retinoic acid (RA; the active metabolite of VitA), has been shown to reduce ΔNp63α expression in HBCs, thereby promoting the differentiation of multipotent Sox2^+^ and Pax6^+^ progenitors via the canonical Wnt signaling pathway in the OE [10,25,26]. Furthermore, the neurophysiological importance of RA signaling in the OE has also been demonstrated in the maintenance and survival of Ascl1^+^ GBCs [27,28].

Within the CNS, RA machineries have been identified in the murine and human brains, especially at the hippocampus and dentate gyrus [29,30,31]. RA increases neurogenesis in the rodent subventricular zone (SVZ)–OB pathway, as demonstrated by increased bromodeoxyuridine-positive (BrdU^+^, proliferating cell marker) cells in SVZ neurospheres and altered cellular migration to the olfactory bulbs [32]. In addition, depletion of doublecortin-positive (DCX^+^, immature neuronal marker) cells in the adult murine dentate gyrus was evident in the retinoid–deficient mouse model, indicating the crucial role of RA in the survival of neural progenitors [33]. 

Olfactory stimulation via ST, as the mainstay of non-pharmaceutical intervention for post-infectious OD, was used to complement the therapeutic effects of oral VitA [14,15,34]. MRI studies in ST-treated patients have demonstrated functional network improvements and structural increase of cortical thickness in the frontal cortex, where the olfactory apparatus is located [19,20,35]. The exact therapeutic mechanism of ST is unknown; however, olfactory stimulations remain essential to the neuro-rehabilitation processes [36]. Reversible and irreversible olfactory occlusion experiments in murine models have shown that olfactory stimuli were contributory to olfactory neurogenesis and glomeruli maturation [37]. In this report, ST was delivered by the novel EPAR diffusers, which utilized ultrasonification to generate aerosolized essential oils for olfactory stimulation. The EPAR diffuser technology enhances the ST experience by facilitating the physiological penetration of aromatic molecules through the olfactory meatus to the OE at the roof of the nasal cavity, therefore providing potent olfactory stimulation and neurosensory rehabilitation.

## 4. Conclusions

In conclusion, structural and functional olfactory defects were identified in patients presenting with prolonged COVID-19-related OD. Preliminary evidence suggested that combination treatment by oral VitA and ST may facilitate the recovery of olfaction, restore OB volumes, and improve functional olfactory network connectivity. The therapeutic efficacy of oral VitA and ST for prolonged COVID-19-related OD should be validated in large scale randomized–controlled trials.

## Figures and Tables

**Figure 1 brainsci-11-00686-f001:**
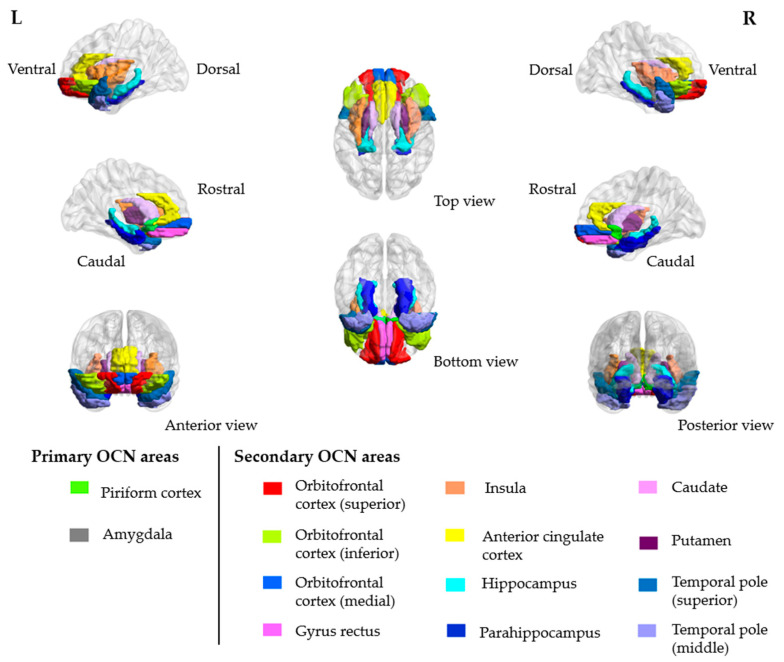
Three-dimensional (3D) representations of the primary and secondary olfactory cortical network (OCN) areas.

**Figure 2 brainsci-11-00686-f002:**
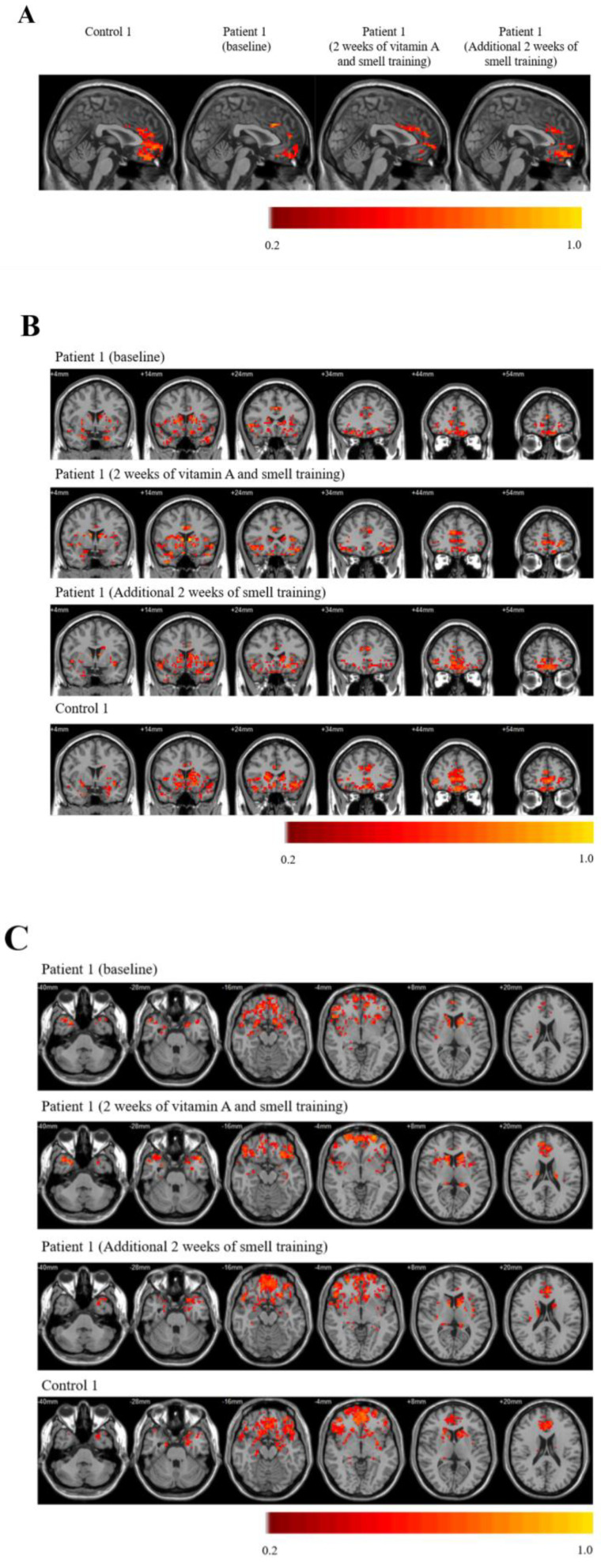
Olfactory network functional connectivity (FC) images using the left caudate as the seed region, as represented in the (**A**) sagittal, (**B**) coronal, and (**C**) axial planes.

## Data Availability

The data presented in this study are available in the manuscript text and Appendix A of this article.

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
