# Peer review of "Neurosensory Rehabilitation and Olfactory Network Recovery in Covid-19-related Olfactory Dysfunction"

_brainsci, 2021, doi:10.3390/brainsci11060686_

Round 1

Reviewer 1 Report

In the manuscript entitled “Neurosensory rehabilitation and olfactory network recovery in COVID-19-related olfactory dysfunction”, Tom Wai-Hin Chung and colleagues provide a preliminary data about structural (olfactory bulb volume reduction) and functional (olfactory network functional connectivity defects) deficits induced by SARS-CoV-2 infection. Additionally, the outcome of the implemented olfactory treatment (OT; a combination of oral vit. A and smell training) is presented.

I find the study very interesting and of potential clinical relevance. However, based on the results only from 1 patient and taking into account the fact that some COVID-19 patients with OD resolve olfactory symptoms within 2 weeks, I would be extra cautious in formulating statements about efficacy of this OT. You will find my detailed comments below.

  1. In the Introduction I miss general information about the percentage of the SARS-CoV-2 infected patients who suffer from olfactory dysfunction (OD). Is there any gender/age/other factor(?) dependence? In how many the infected patients OD is transient (according to some reports olfactory symptoms may resolve spontaneously within approx. 2 weeks see: 2020;323(24):2512-2514 doi:10.1001/jama.2020.8391) and how many of them develop the permanent post-infectious OD?
  2. Goals of the study should be formulated.
  3. What are the chances for a spontaneous olfactory recovery in the patient/s that have received OT (e.g. vit. A + olfactory stimulation)?
  4. Table 1 – To state if COVID-19 patients/controls presented certain symptoms “1” and “0/Nil” (the difference between them is unclear) were used. I suggest using simply yes/no (or lack of data).
  5. Figure 1 – The font in the fig. legend with names of the brain regions is too small – they are barely readable. Also, the exact descriptions from which perspective we see each brain would be really helpful.
  6. L123-125 language correction of this sentence should be done.
  7. What is the general meaning of the OD? Does it have any prognostic value in patients with SARS-CoV-2 infection? Do patients with the post-infectious OD are more prone to develop other CNS disorders? The Authors should discuss it.

Reviewer 2 Report

Manuscript titled "Neurosensory rehabilitation and olfactory network recovery in 2 COVID-19-related olfactory dysfunction" submitted by Chung et al., to MDPI - Brain Sciences as a case report is interesting. Authors report a case study of COVID-19 patients presented with symptoms including olfactory  dysfunction - anosmia and microsmia. Due to the current COVID-19 pandemic concerns, this case study will be of significant interest among readers. However, the manuscript needs significant improvement according to my suggestions below;

1) In the introduction, include findings from previous studies that used VitA to improve olfaction in general, OE regeneration etc.

2) Rearrange Table 1 to segregate patients from controls (especially C1) clearly. Also check the manuscript for typos - "Anosmia" not Ansomia.

3) In Figure 1, can you add planes and coordinates? Dorsal - Ventral, Rostral - Caudal, Superior - Inferior, Anterior - Posterior. Also top view? or bottom view? Figure 1 can be divided to more subpanels.

4) It is not clear if the how much patients underwent VitA and olfactory treatment (OT). Did controls underwent any OT?  Did all controls have MRI brain scans? What are the clinical outcomes of Patient 2 without VitA/OT?

5) Your conclusions does not match your findings other than you found olfactory defects in COVID-19 patients. You can only speculate that VitA and OT had any effect because you have not evaluated enough number of patients who recovered from the olfactory dysfunctions associated with COVID-19 without treatments like VitA/ OT. It is partially acceptable, if you were to report only Patient 1 as a case report.

6) Supplementary data is not available. It is very difficult to comprehend the manuscript without the supplementary information.

Round 2

Reviewer 2 Report

Authors Chung et al., submitted a revised version of their manuscript titled "Neurosensory rehabilitation and olfactory network recovery in 2 COVID-19-related olfactory dysfunction" to MDPI - Brain Sciences as a case report. 

I accept the changes made by the authors.